# Full Lead Service Line Replacement: A Case Study of Equity in Environmental Remediation

**Karen J. Baehler** [1,*] , **Marquise McGraw** [1] , **Michele J. Aquino** [1] , **Ryan Heslin** [1] , **Lindsay McCormick** [2] **and Tom Neltner** [2]

1   School of Public Affairs, American University, Washington, DC 20016, USA; mcgrawm@american.edu (M.M.); ma6936a@american.edu (M.J.A.); rh3788a@student.american.edu (R.H.)

2   Environmental Defense Fund, Washington, DC 20009, USA; lmccormick@edf.org (L.M.); tneltner@edf.org (T.N.)

\*   Correspondence: baehler@american.edu; Tel.: +1-202-885-6072

**Abstract:** In the U.S., approximately 9.3 million lead service lines (LSLs) account for most lead contamination of drinking water. As the commitment to replace LSLs with safer materials grows, empirical evidence is needed to understand which households are benefitting most from current replacement practices. This exploratory study analyzes factors predictive of whether an LSL was replaced fully (from water main to premise) or partially (only the portion on public property). Conventional ordinary least squares, negative binomial, and geographically weighted regression models are used to test the hypothesis that full lead service line replacements (LSLRs) were less common in lower-income, higher-minority neighborhoods under a cost-sharing program design in Washington, D.C. between 2009 and 2018. The study finds supportive evidence that household income is a major predictor of full replacement prevalence, with race also showing significance in some analyses. These findings highlight the need for further research into patterns of full versus partial LSLR across the U.S. and may inform future decisions about LSLR policy and program design.

**Keywords:** lead contamination; drinking water; lead service line replacement; health equity; environmental justice; environmental policy; environmental remediation; water utilities

## 1. Introduction

Lead water pipes, also known as lead service lines (LSLs), were widely used throughout the United States (U.S.) until the 1980s. A service line refers to the span of plumbing that connects a building with the public water main under the street. When any portion of this service line is composed of lead pipe it is termed an LSL. Lead materials were historically used because of favorable physical and chemical properties, specifically pliability and relatively low corrosiveness [1]. It is now widely accepted that this plumbing material choice is a source of lead in drinking water.

The harms associated with lead exposure are widely known and well documented [2]. There is no known safe level of lead exposure, and children are particularly vulnerable due to their developing brains and bodies [3]. A vast and continuously growing literature indicates that negative effects in children include anemia, hearing impairment, slower growth, lower IQ, hyperactivity, and behavioral and learning problems [2,3]. In adults, lead exposure has been linked to cardiovascular, renal, hepatic, immunological, and reproductive problems [2,4–8]. Maternal lead exposure creates risk of fetal brain damage in utero [2].

Water is an important source of lead exposure, especially for formula-fed infants who may consume contaminated water through reconstituted formula [9]. When present, LSLs are the largest source of lead in water, contributing 50–75 percent of the lead [10]. Contamination risk is complex and tied to multiple, interacting factors including source water chemistry (pH or degree of acidity), water temperature, flow rates, varieties of materials

found in service lines and premise plumbing, and other system features. Water treatment-based measures to control corrosion have played a vital role in reducing lead release into drinking water across the U.S. for decades. Nonetheless, research and experience have shown that LSLs can leach soluble lead into drinking water and unpredictably release lead particulates even in systems with corrosion control, especially when service lines are disturbed or water treatment is disrupted [11,12]. Although recent empirical work indicates that water main replacements do not increase lead levels in drinking water under all circumstances [13], water main work cannot be ruled out as a risk alongside other physical disturbances such as road repairs, building construction, and hydraulic shocks such as sudden flow changes [10,11,14].

The goal of eliminating LSLs from U.S. water distribution systems has gained significant momentum in recent years. In 2015, the National Drinking Water Advisory Committee (NDWAC) exhorted water systems to "work with their customers to implement full replacement of all lead service lines in their service areas" [15] (p. 16). In a 2017 policy statement following the widely publicized tragedy in Flint, Michigan, the American Water Works Association called on "communities to develop a lead reduction strategy that includes identifying and removing all lead service lines over time" and replacing them with non-lead materials all the way from the water main to the building [16] (p. 1).

Since 2016, 17 states representing a total of 4–5 million LSLs have created proactive policies to support full lead service line replacement (LSLR) programs within their boundaries according to a recent count [17]. Those state-level policies are in turn bolstering the efforts of towns and cities, 115 of which have set a goal of eliminating LSLs [17].

Progress toward the goal varies from place to place. As of early 2021, nine communities had achieved LSL elimination and the cities of Flint, Michigan and Newark, New Jersey were on track to finish replacing their full inventory of LSLs in 2021 [17–19]. By contrast, the Mayor of Chicago in late 2020 announced a new plan to "chip away" at the city's 400,000 LSLs in a process expected to stretch "over several decades," in the mayor's words [20] (p. 1). Nationwide, an estimated 9.3 million LSLs await replacement at a current cost range of $3953–$6024 per line, according to the U.S. Environmental Protection Agency (EPA) [21] (Exhibit 5-9). Other estimates range from 6.1 to 12.8 million LSLs [22,23]. The job of fully replacing all LSLs is likely to take decades under even the best-case scenario. Pandemic-related fiscal pressure on state and local government budgets in 2021 and beyond will likely stretch target dates even further into the future. Notably, the Biden administration has been vocal about LSLR as a U.S. national infrastructure priority [24].

Protracted timeframes for LSLR obligate policymakers to think about whose LSLs will be fully removed and replaced sooner rather than later, and how to fund the billion-plus dollars required to complete the work. Full replacement refers to removal of the entire length of an LSL from water main to premise and replacement with non-lead material, including segments of the service line located on both public and private property. Partial LSL replacement involves removing and replacing only the portion of pipe on public land (between the water main and the property line) and connecting the new non-lead pipe on the public side to the old lead pipe on the private side (from the property line to the building). Full replacement is the correct policy target because only full replacement can eliminate service lines as sources of lead exposure risk. The NDWAC definitively states that "[l]ead-bearing plumbing materials in contact with drinking water pose a risk at all times (not just when there is a lead action level (LAL) exceedance)" [15] (p. 7). The AWWA likewise warns of continued risks to public health "as long as there is lead in contact with drinking water" [16] (p. 1).

Partial replacement is not a second-best alternative to full replacement and may be worse than doing nothing. In some cases, partial replacements have been shown to accelerate lead release due to galvanic corrosion and/or disturbance of pipe coatings that normally protect against lead leaching [25,26]. Dissolved and particulate lead may spike to very high levels following a partial replacement, and higher lead levels may persist in the drinking water for months [27–29]. Even if those risks are controlled through flushing,

corrosion control treatment at the water plant, and other measures (e.g., publicly supplied water filters), such efforts have limits. Allowing any amount of lead pipe to remain in the ground poses future risks of lead release. In a December 2021 announcement, the EPA reaffirmed this key point: "Partial LSLRs can cause short-term elevation of lead concentrations in drinking water and further extend lead health risk from service lines because a portion of the lead line remains in service. EPA strongly discourages water systems from conducting partial LSLR. EPA recommends systems proactively implement full LSLR programs" [30] (p. 71582).

Which American households will experience the highest cumulative risk of lead exposure while thousands of water systems work their way through roughly 9 million remaining LSLRs? The environmental justice and public health literatures provide reasons to worry that historically disadvantaged populations will bear the brunt of the burden. In the U.S., measures of household financial means and markers of racial and ethnic identity have proven to be statistically significant predictors of multiple environmental and health outcomes, including drinking water quality and lead exposure from all sources [31–34]. Specific to the water sector, a recent EPA-funded environmental-justice analysis estimated disproportionately high baseline levels of lead exposure risk for minority and low-income households due to the higher probability of these populations occupying dwellings built in decades when LSLs were most often installed [35]. Likewise, the U.S. Government Accountability Office found higher concentrations of LSLs in neighborhoods with more markers of vulnerability, including higher poverty and unemployment rates, larger minority populations, more single female-headed households, more renters, and lower educational attainment [36].

Black Americans on average face especially high risks due to multiple, interacting disadvantages with deep historical roots. The median level of household wealth in 2019 was 7.8 times larger for White households than Black households ($188,220 compared with $24,100), and average levels of wealth were even more skewed than medians [37]. Well-known research by Raj Chetty and colleagues [38] documents lower rates of upward mobility and higher rates of downward mobility for Black Americans compared to Whites and Hispanic Americans. According to one study of a notable environmental outcome— proximity to facilities that emit harmful particulate matter—non-Whites were found to experience 1.28 times higher burden of exposure than the general population, and among non-Whites, Blacks had 1.54 times higher burden [39]. With respect to lead exposure specifically, a review of multiple studies found the highest mean blood lead levels among Black Americans generally; and in studies reporting blood lead ranges, Black children were more likely than other groups to experience elevated levels [34].

Given these baseline facts, even if future full LSLRs occur at identical rates for minority and non-minority households, and at identical rates for all households across the income distribution, population groups with higher baseline risk (especially Black Americans) will continue to bear a higher level of risk until all LSLs are eliminated. Closing the race- and income-based gap in baseline exposure risk would require fast tracking full LSLR for low-income households and households of color.

Instead of either fast tracking or equal pacing, however, actual rates of full LSLR are likely to be slower for low-income, minority households in the future for as long as current cost-sharing arrangements remain in place. These local arrangements, which are commonly used throughout the country, require property owners to pay to replace LSL components located on private property at an estimated cost of $2514 to $3929 per home [21] (Exhibit 5-9). "Since the LSLR is expensive," cost-sharing requirements are likely to produce fewer full replacements for low-income households because "the customer's willingness to share costs will depend on the household's ability-to-pay" [35] (p. 14). As a result, partial replacements "may be unavoidable if low-income households are unable to afford the cost [of full replacement] and the system or other agencies do not subsidize LSLR for low-income households" [35] (pp. 14–15).

Some local governments and utilities have begun to address the equity issues associated with LSLR proactively by fully funding full LSLR, but this practice is still rare [17,40]. The infrastructure funding bill passed by the U.S. Congress in November 2021 will make $15 billion available nationwide—roughly one-third of the total estimated cost of LSL elimination. Thus, the EPA expects cost-sharing to be the norm going forward in the context of goal-based, non-mandated LSLR initiatives. Specifically, EPA's formula for calculating the per-unit cost of a non-mandated full replacement rests on the assumption "that CWSs [Community Water Systems] will only incur costs for the utility side of the LSLR, and that customers will pay for their portion to achieve full replacements" [21] (pp. 5–188). The new federal funding is likely to reduce local water systems' reliance on cost-sharing but will not end it.

In the absence of prohibitions on partial replacements, LSLR programs that continue to employ cost-sharing may be at risk of perpetuating existing environmental and public health injustices tied to income and race. If partial replacements are found to be more common among lower-income households, including disproportionate numbers of Black households, the risks associated with partial LSLR will load on top of existing disparities in environmental and health burdens. Faced with this conspicuous social equity risk, it is useful to gain a better understanding of the factors that may obstruct full replacement, especially ability to pay.

This article is the first to explore social equity risk associated with LSLR empirically using data on actual full and partial replacements. We do so by modeling patterns of full versus partial LSLR prevalence at the neighborhood level using administrative data on all LSLRs conducted by or reported to the utility that serves Washington, D.C. between 2009 and 2018. The results of the exploratory models point to significant and positive correlations between neighborhood income and prevalence of environmental remediation in the form of full LSLR. Neighborhood racial characteristics also appear to have some predictive power, with larger percentages of Black householders associated with lower probability of full replacement at more aggregated area levels. The evidence presented here, though not definitive, provides impetus for further research as well as regular monitoring of who is experiencing full versus partial—effective versus potentially harmful—remediation of LSL risk.

## 2. Materials and Methods

### 2.1. Hypothesis and Approach

The EPA's environmental justice analysis of LSLR does not state a hypothesis outright, but it implies the following hypothesis: under cost-sharing arrangements common to many water systems, households with greater financial means, many of whom are White, are more likely to receive a full rather than partial LSL replacement compared to lower-income, minority households (and vice versa) [35]. The environmental justice and public health literatures strongly support this hypothesis, as does common sense, but no empirical studies have attempted to test it until now.

The study team constructed a unique database with information about date and type of LSLR at the premise level for 3419 full and partial LSLRs completed between 2009 and 2018 plus information about household income and Black race at the neighborhood level. Ideally such a model would employ premise-level data for all variables, which in turn would require access to Census Bureau data on household characteristics by street address. The study team did not have such access and therefore used area-based measures of income and racial composition, as have many previous studies of lead exposure [41–46]. Under the approach adopted for this study, area-level demographic information estimates a given household's probability of having a particular income profile and racial/ethnic identity.

### 2.2. Case Study Background

The tests reported in this article are based on an exploratory effort to model neighborhood prevalence of full versus partial replacements as a function of household characteris-

tics in Washington, D.C. under a cost-sharing regime that operated via two programs during the study period 2009–2018. Under the first, which we refer to as customer-initiated (CI) LSLR, the homeowners either organized the replacements themselves or took the initiative to contact DC Water (the local utility serving Washington, D.C.) to send a crew to conduct a full replacement. All customer-initiated LSLRs were full replacements, by definition.

The second program, which we refer to as utility-initiated (UI) LSLR, refers to situations in which homeowners were offered the opportunity to fully replace their LSLs at the same time the utility was undertaking other infrastructure work on their street as part of a planned project (capital improvement) or in response to an emergency such as a leaking water main. In advance of infrastructure work on a block, the utility provided information to property owners about the importance of fully replacing LSLs, the convenience of scheduling the full LSLR through DC Water's contractors in conjunction with other infrastructure work, and the average price range for the customer's portion of the bill. At premises where property owners did not respond to DC Water's outreach, the utility's standard practice was to conduct a partial replacement. Partials did not require customer approval. The utility covered all costs for partials.

In both programs during the study period, property owners had to agree to pay for replacing the LSL portion on private property to receive a full replacement, and all customers were treated equally in this respect regardless of underlying differences in households' financial capacities. The customer's share of the cost of a full replacement averaged $2500 per home under the UI arrangement or $3200 per home for a CI. The 28-percent difference in price between UI and CI reflects efficiencies associated with piggybacking a full replacement on other projects: when the utility is already opening the street, the marginal cost of one more replacement job is lower. By contrast, customer-initiated replacements require a special, scheduled trip to the property plus excavation, which creates additional costs.

As shown in Table 1, nearly two-thirds of all replacements during the study period were customer-initiated.

**Table 1.** Count of LSLRs by program and type, Washington, D.C. 2009–2018.

| Type of LSLR | Program | | | | Total CI + UI |
|---|---|---|---|---|---|
| | Customer-Initiated LSLRs (CI) | Utility-Initiated LSLRs (UI) | | | |
| | | Capital Improvement Projects | Emergency Repairs | All UI | |
| Fulls | 1356 | 805 | 86 | 891 | 2247 (65.7%) |
| Partials | 0 | 815 | 357 | 1172 | 1172 (34.3%) |
| Total (%) | 1356 (40%) | 1620 | 443 | 2063 (60%) | 3419 (100%) |

The Washington, D.C. City Council adopted and approved funding for a new approach to LSL replacement in 2019. The new scheme focused on remedying past partial replacements at the city's expense and ensuring more equal access to full replacement in the future through city funding of full UI replacements. Our study period closes in 2018 to correspond with the end of the earlier cost-sharing scheme. Future research will compare patterns of full versus partial replacement before and after the 2019 policy change.

### 2.3. Dependent Variables and Data Sources

American University signed a Memorandum of Understanding with DC Water in April 2019 giving the AU research team access to a dataset with basic information about all LSLRs conducted in Washington, D.C. between 2009 and 2018, inclusive. The study period

was chosen to capture replacement patterns associated with cost-sharing for full LSLR—the policy in place in DC Water's service area prior to October 2019—and to maximize data quality, which improved starting in 2008 due to new administrative protocols at the utility.

The analyses reported here seek to identify influences on two dependent (outcome) variables at the neighborhood level for the study period: (1) raw count of full LSLRs and (2) percentage of total LSLRs in which the service lines were fully replaced (full LSLR/total LSLR), sometimes referred to in this article as the rate of full replacement. Higher values for either dependent variable represent more desirable health and environmental outcomes because only full removal and replacement of the LSL ensures complete and permanent protection from service-line-related lead exposure. Because partial replacements are inferior to full replacements, as explained previously, the study design does not use combined full and partial replacements as an outcome variable.

*2.4. Area Levels*

Neighborhood patterns are examined at two levels: census tract (using 2010 census tract boundaries) and ward. The latter refers to the city's 8 primary political subdivisions. In order to match each LSLR to a census tract and ward, 67 duplicate or incomplete records were removed from the DC Water database, leaving a final set of 3419 addresses where replacements of any type had occurred during the study period. Many of those duplicate records consisted of a partial replacement followed by a full replacement. In the 46 cases in which a full replacement occurred within two years of an earlier partial replacement, the premise was coded once as full, and the other observation was dropped. If more than two years elapsed between the partial and full replacements (3 cases), the premise was counted twice—once as partial and once as full—to acknowledge the extended risk faced by residents following the partial replacement and before its full remediation. Two customer-initiated replacements identified as partials also were removed because all customer-initiated replacements should be full replacements: these data points likely represented administrative errors.

The data were then geocoded, with every LSLR address assigned to a census tract and ward using ArcGIS. Out of 179 census tracts in Washington, D.C. as defined by the 2010 Census, six census tracts were dropped from this study due to lack of residential housing of the type likely to have lead service lines. These include a military zone, the White House and National Mall, and census tracts consisting entirely of hospitals, a prison, a sports stadium, and two universities.

Replacements of any type were unevenly distributed across all eight wards (Table 2) and 141 census tracts (Table 3); 32 census tracts out of 173 used in this study had zero LSLRs.

**Table 2.** Count of LSLRs by ward and type, 2009–2018.

| Ward | Type of LSLR | | Total |
|:---:|:---:|:---:|:---:|
| | **Full** | **Partial** | |
| 1 | 234 | 173 | 407 |
| 2 | 151 | 98 | 249 |
| 3 | 297 | 59 | 356 |
| 4 | 461 | 120 | 581 |
| 5 | 363 | 263 | 626 |
| 6 | 623 | 284 | 907 |
| 7 | 28 | 37 | 65 |
| 8 | 90 | 138 | 228 |
| DC Total (Total %) | 2247 (65.7%) | 1172 (34.3%) | 3419 (100%) |

**Table 3.** Descriptive statistics for LSLRs at census-tract level, by type, 2009–2018.

| | Type of LSLR | | Total |
|---|---|---|---|
| | **Full** | **Partial** | |
| Smallest number of LSLRs in a single census tract | 0 | 0 | 0 |
| Largest number of LSLRs in a single census tract | 95 | 92 | 160 |
| Median number of replacements for all census tracts | 9 | 3 | 14 |
| Mean number of replacements for all census tracts | 15.94 | 8.31 | 24.25 |
| Standard Deviation from the mean | 18.70 | 13.05 | 28.96 |

*2.5. Independent Variables and Data Sources*

　　To test for health equity and environmental justice impacts, the outcome variables (number of fulls and percentage full) were correlated with demographic characteristics—race and income, specifically—to determine if full replacements were more likely in D.C.'s more advantaged neighborhoods during the study period and vice versa, as hypothesized.

　　Race and income variables were sourced from the U.S. Census Bureau's American Community Survey (ACS). The ACS publicly available files report 5-year averages at the census-tract level to ensure adequate sample sizes. ACS estimates from 2013 (incorporating annual survey results from 2009 to 2013) and 2018 (incorporating annual survey results from 2014 to 2018) were averaged to cover all relevant study years and avoid double counting any years. The following ACS variables were used at the census tract and ward levels:

- Median household income is a measure of capacity to pay for full replacements in systems where customers are expected to pay for work done on private property. Therefore, a positive correlation between median income and either outcome variable is expected. In other words, demand for full replacements is expected to be elastic with respect to income, recognizing that these variables represent demand at a neighborhood level rather than individual consumer level.
- Percent Black householders is expected to be negatively correlated with the outcome variables because it captures another potential source of systemic social disadvantage, beyond income, that could function as an obstacle to accessing full LSLR. Householder refers to an adult in whose name the housing unit is either owned or rented and maintained (or any adult if no householder is present). The Black race variable was employed because of Washington, D.C.'s long history of Black–White hypersegregation [47,48]. In 2016, midway through this project's study period, D.C. had an index of dissimilarity score (i.e., segregation index score) of 71 out of 100, which means 71 percent of D.C.'s Black residents would have to move to different census tracts to produce a pattern of racial distribution at the census-tract level that mirrors D.C.'s overall racial composition [49]. Scores above 60 on this index are considered very high [48]. (Percent Hispanic householders was not significant in any of the models and was therefore dropped.)

　　As Tables 4 and 5 demonstrate, demographic characteristics varied dramatically across both wards and census tracts during the study period.

**Table 4.** Descriptive statistics for demographic variables at ward level.

| Ward | Median Household Income in $ (Average of 5-Year Averages for 2013 and 2018) | Percent Black Householders (Average of 5-Year Averages for 2013 and 2018) | SDI Score (2015) |
|------|------|------|------|
| 1 | 89,557 | 24.10 | 68.4 |
| 2 | 103,734 | 9.24 | 39.9 |
| 3 | 130,696 | 5.50 | 29.2 |
| 4 | 84,410 | 48.53 | 60.0 |
| 5 | 63,060 | 57.55 | 63.9 |
| 6 | 104,873 | 26.12 | 43.5 |
| 7 | 47,691 | 70.08 | 87.6 |
| 8 | 30,527 | 73.79 | 95.2 |

**Table 5.** Descriptive statistics for demographic variables at census-tract level.

| All Census Tracts | Median Household Income in $ (Average of 5-Year Averages for 2013 and 2018) | Percent Black Householders (Average of 5-Year Averages for 2013 and 2018) |
|------|------|------|
| Lowest value in a single census tract | 14,572 | 1.4 |
| Highest value in a single census tract | 203,382 | 99.4 |
| Median value for all census tracts | 75,703 | 48.3 |

An additional independent variable—count of residences built before 1950 (housing vintage)—was included to control for the potential total pool of premises with lead service lines in each census tract. Previous research has shown that housing age serves as a reasonably reliable predictor of the presence of LSL in some locations [50]. In Washington, D.C., most lead service lines were installed prior to 1936, according to utility officials with whom we corresponded. Lead installations increased 1941–1946 due to World War II supply demands on copper and then declined to less than 5 per year in 1948. Sporadic installations continued until 1977. Given this background, 1950 looks like the correct cut-off year for Washington, D.C.

*2.6. Ward-Level Analysis*

Analysis of health equity and environmental justice dimensions began with a preliminary exploration of patterns at the ward level. The study team compared rates of full replacement (number of full LSLRs/total LSLRs) for wards with the highest and lowest markers of social vulnerability based on race and income. In addition, rates of full replacement for each of the eight wards were correlated separately with race, income, and a measure of social deprivation known as the Social Deprivation Index on a simple bivariate basis: those results are reported in Appendix A.

Although ward-level analysis provides too few observations (n = 8) for regression modeling, it offers an opportunity to look at how patterns of LSLR vary across areas that represent meaningful social and political divisions in Washington, D.C. Historical patterns of residential segregation in the nation's capital have tended to follow ward boundaries, and as a result, reports on socio-economic conditions in D.C. often present findings at the ward level [51–56].

In addition, ward-level analysis avoids the problem of census tracts with small numbers of replacements. In those census tracts the addition or subtraction of just one or two

full LSLRs can dramatically change the overall rate of full replacement for that tract and thus exaggerate differences between tracts. The small numbers problem does not occur at the ward level thanks to large numbers of full replacements in each of the eight wards compared to each of 179 census tracts.

*2.7. Census-Tract-Level Regression Analysis*

Measuring the strength of this study's hypothesized relationships requires regression analysis at the census tract level where a larger "n" is available compared to the ward level. The study team ran two different types of regression as a check-and-balance. Negative binomial regression (NBR) is considered a good choice for models in which the dependent variable is a count variable and when the data comprising that variable are over-dispersed, meaning their variance is greater than their mean [57]. Census tracts with zero full LSLRs contribute to over-dispersion in our model's count-type dependent variable. Tests of over-dispersion compared conditional means and variances for each variable and found variances to be larger than means, which supports use of the NBR model. Additional diagnostics, including a Q–Q plot of full replacements (the dependent variable), also support the choice of NBR.

Ordinary least squares (OLS) regression also was applied to establish baseline results and because many readers will be most familiar with this approach and find it easiest to interpret. Both types of models were run in R and Stata with robust standard errors. Median household income and percentage of Black householders, described earlier, comprise the explanatory variables in the main models. They are included to test the hypothesis that areas with higher incomes and smaller concentrations of Black households will experience more full replacements.

Regression models were run using the number of full replacements in each census tract as the dependent (outcome) variable rather than the rate of full replacements. Using the raw count of full LSLRs avoids the small numbers problem described earlier and enables the model to account for differences in the scale of the LSLR intervention in each neighborhood. Scale is important because a neighborhood could have many full replacements but a small percentage of fulls if their total number of LSLRs (partials and fulls) is large. Likewise, a neighborhood could have a high percentage of fulls but a small number of full replacements if their total number of LSLRs is small.

It is worth noting that the housing vintage control variable serves an additional purpose in the models. It functions roughly like the denominator of a ratio with the dependent variable (count of fulls) as the numerator. If LSLs were being fully replaced at exactly equal rates in all parts of the city, and if pre-1950 housing is an accurate estimator of LSL presence, then the correlation between count of full replacements and count of pre-1950 housing units would be close to perfect (=1). In the negative binomial model (see below), the pre-1950 housing count variable functions as the exposure term, which measures the maximum number of possible events (in this case, LSLRs) in each census tract.

The resulting formal model can be written: $FC_i = \beta_0 + \beta_1 R_i + \beta_2 I_i + \beta_3 V_i + \beta_4 W_i + \varepsilon_i$

- $FC$ = Count of full replacements in census tract $i$
- $R$ = Race, measured by percentage of householders identifying as Black/African American in census tract $i$
- $I$ = Median household income in census tract $i$
- $V$ = Housing vintage, measured by number of housing units in census tract $i$
- built prior to 1950
- $W$ = Ward fixed effects (binary indicator/dummy variables for each ward).

Ward fixed effects were included in the conventional OLS and NBR models to account for persistent and significant differences between wards on many indicators of socio-economic status and public health. As discussed in Section 2.5, ward boundaries capture important features of life in Washington, D.C., including patterns of residential segregation, which may influence the probability of having a full LSLR. Ward fixed effects offer a complementary strategy to the geographically weighted (GWR) models described below

by accounting for unobserved, but potentially germane, differences between wards that also manifest at the census-tract level.

We ran all conventional (pre-GWR) models with and without fixed effects. The models with ward fixed effects typically had higher overall significance (adjusted R-squared scores and equivalents) and were more "conservative" than the models without ward fixed effects, meaning the explanatory power of the main independent variables—race and income—was lower in the fixed-effects models. The fixed-effects conventional models offer a tougher test of this study's hypothesis and provide one approach to accounting for geographic variation, and are therefore the preferred model specification pre-GWR.

Conventional OLS models also were run using the natural logs of all variables, which generates a more normal distribution of the underlying values for ease of interpretation. In the logged models, the regression coefficient on the median household income variable provides a proxy for income elasticity of demand. Those results are reported in Appendix B (for all full LSLR) and Appendix C (disaggregated by program type). The log-transformed models in Appendix B report an "n" of 129 because the log of zero is not a real number, and therefore, census tracts with zero full LSLRs drop out. In Appendix C, the "n's" are smaller still because they include only census tracts containing LSLRs associated with each program type.

### 2.8. Spatial Autocorrelation, Spatial Non-Stationarity, and Geographically Weighted Regression (GWR)

Washington, D.C.'s high level of residential segregation means that relevant features of census tracts may be systematically patterned rather than random. This phenomenon, known as spatial autocorrelation or spatial dependency, has been found in other studies of environmental justice [58,59]. Moran's I test produced a value of 0.070 with standard deviation of 0.008 and a *p*-value of zero for our dependent variable (full LSLRs). We therefore reject the null hypothesis of no spatial autocorrelation in this dataset [60]. This test tells us that census-tract-level values for full replacement are not randomly distributed in D.C. and alerts us to the possibility that our conventional OLS and NBR results may therefore be biased.

To better understand how geography enters our results, as well as to mitigate bias caused by spatial autocorrelation, we ran geographically weighted regression (GWR) models. GWR has the potential to correct for spatial autocorrelation by taking distance between census tracts into account when calculating parameter estimates. It does so by conducting a series of regressions, one for each observation in the dataset—in this case, one for each census tract. In each separate regression, the focal census tract's dependent and independent variable values are given full weight while the values for the other census tracts are weighted according to their distance from the focal census tract. More distant census tracts' values are weighted less. The rate at which weights diminish as distance increases (the distance decay function) depends on a setting known as the model's bandwidth.

In addition to addressing spatial autocorrelation, GWR also enables exploration of non-stationarity, a phenomenon in which the statistical relationships between independent and dependent variables vary from place to place within the geographic area. In this case, GWR allows us to inquire about whether the relationships between race, income, and rates of full LSLR might be different in different parts of D.C.

We ran both OLS and NBR models using the GWR package in Stata, which incorporates an algorithm that sets bandwidths automatically. For each analysis, the first step involves testing the bandwidth's significance, which signals whether the GWR model is more appropriate than the standard OLS or NBR model for this data. The second step tests each variable in the model for non-stationarity, which involves comparing the GWR results against data obtained from a Monte Carlo simulation with 1000 observations.

## 3. Results

### 3.1. Ward-Level Results

The ward-level analysis provides preliminary evidence of a meaningful connection between spatial variation in full LSLR participation and spatial variation in population characteristics at the neighborhood level in Washington D.C. Table 6 displays the variation in rates of full replacement (full LSLRs/all LSLRs) for the highest- and lowest-income wards and wards with the largest and smallest percentages of Black householders. The middle two rows show dramatically higher rates of full replacement in wards with the highest incomes and smallest percentages of Black householders, compared to those with the lowest incomes and highest percentages of Black householders, for both total LSLRs and LSLRs associated with capital improvement projects.

**Table 6.** Ward-level participation in full LSLR by program, 2009–2018 (omits customer-initiated LSLRs because they are all full replacements; none are partials).

| Demographic Information | | Full LSLR as a Percent of Total LSLR | | | |
| | | TOTAL LSLR Utility-Initiated (UI) + Customer-Initiated (CI) | Utility-Initiated (UI) | | Wards |
| Variable | Quartile | | Capital Improvement Projects | Emergency Repairs | |
|---|---|---|---|---|---|
| Percent Black Householders | Top 2 wards, 4th quartile, (highest concentration) | 40 | 33 | 26 | 7 and 8 |
| | Bottom 2 wards, 1st quartile (lowest concentration) | 74 | 63 | 22 | 2 and 3 |
| Median Household Income | Top 2 wards, 4th quartile (highest income) | 73 | 60 | 15 | 3 and 6 |
| | Bottom 2 wards, 1st quartile (lowest income) | 40 | 33 | 26 | 7 and 8 |

In each case (total and capital projects), the difference in full replacement rates between the least and most advantaged quartiles of wards is 30 percentage points or more. By contrast, rates of full replacement when LSLRs are undertaken as part of emergency work on water infrastructure do not vary much across quartiles: no connection between percentage of full LSLRs and demographic characteristics is evident in the case of emergency replacements. Table 6 does not include separate calculations for customer-initiated replacements because those replacements are all, by definition, fulls rather than partials: therefore, percentage full cannot be calculated. Percentage full can be calculated for utility-initiated and customer-initiated LSLRs combined: that aggregated outcome measure captures the total effect of the utility's total LSLR activity on city residents.

Because all households in this database that did not receive a full replacement received a partial replacement, the full replacement rate in each area is the exact inverse of the partial replacement rate. Table 6's results therefore suggest that children and families living in D.C.'s most socially vulnerable wards during the study period were more likely to receive an inferior environmental remedy (a partial replacement) compared with their fellow Washingtonians in less socially vulnerable wards. Appendix A reports the results of ward-level bivariate correlations for rates of full replacement with each of the demographic variables. Those statistically significant correlations offer further circumstantial evidence of a potentially consequential relationship at the ward level between demographic characteristics and the probability of receiving a full LSLR rather than a partial LSLR.

Though preliminary, the results in Table 6 and Appendix A strengthen the case for adding ward fixed effects to the regression models and exploring other aspects of spatial patterning.

### 3.2. Results for Census Tracts Grouped by Quartile

Figures 1 and 2 provide further reasons to suspect a relationship between prevalence of full replacements and demographic characteristics at the area level. These simple visuals organize Washington, D.C. census tracts by quartiles based on percentage of householders identifying as Black/African American (Figure 1) and median household income (Figure 2). Height of the bar for each quartile of census tracts corresponds with the total count of full LSLRs that occurred within that group of census tracts. Figure 1 shows a sharp difference between the fourth quartile and all others, which suggests that race-based inequities in full LSLR participation may be skewed toward the neighborhoods with highest concentrations of Black householders. Figure 2 shows a steady stair-step pattern, which points to a potential linear correlation between income and full LSLR participation.

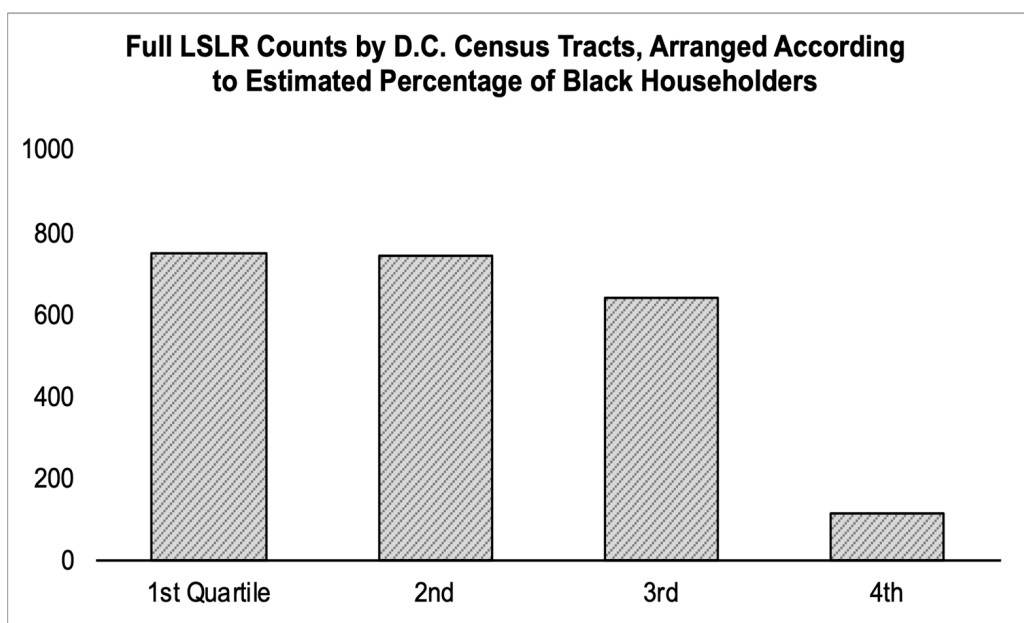

**Figure 1.** Full LSLR counts (totals) are notably smaller in the quartile of census tracts with largest percentages of Black householders.

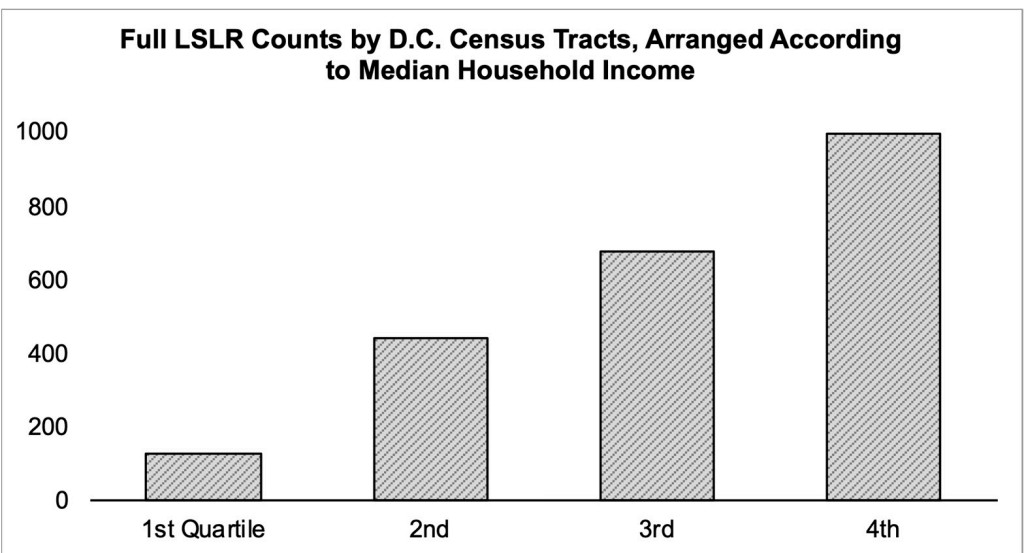

**Figure 2.** Full LSLR counts (totals) increase steadily across census-tract income quartiles.

### 3.3. Census-Tract-Level Regression Results

Regression analysis at the census-tract level allows closer examination of the ward-level findings to observe which independent variables (IVs) appear to have the most explanatory power. Regression also helps assess whether the relationships displayed in Table 6 and Appendix A hold at a more disaggregated area level.

Tables 7 and 8 below summarize results of the regression models. Models 7.1–7.3 (in Table 7) and models 8.1–8.3 (in Table 8) are identical except that the first group was run using OLS and the second was run using negative binomial regression.

**Table 7.** Association of full LSLR with race and income characteristics at census-tract level: OLS model results with robust standard errors (in parentheses).

| Independent Variable | Model 7.1 | Model 7.2 | Model 7.3 |
|---|---|---|---|
| Percent Black Householders | −0.315 *** (0.108) | −0.087 (0.126) | −0.085 (0.120) |
| Median Household Income ($1000s) | | 0.175 *** (0.054) | 0.134 ** (0.052) |
| Vintage Housing Stock Count | | | 0.013 *** (0.003) |
| D.C. Ward Fixed Effects | Yes | Yes | Yes |
| Observations | 173 | 173 | 173 |
| Adjusted $R^2$ | 0.175 | 0.220 | 0.302 |

** $p < 0.05$; *** $p < 0.01$.

**Table 8.** Association of full LSLR with race and income characteristics at census-tract level: negative binomial regression (NBR) model results with robust standard errors (in parentheses) and incident rate ratios (IRR).

| Independent Variable | Model 8.1 | | Model 8.2 | | Model 8.3 | |
|---|---|---|---|---|---|---|
| | Coefficient | IRR | Coefficient | IRR | Coefficient | IRR |
| Percent Black Householders | −0.028 *** (0.009) | 0.972 | −0.007 (0.010) | 0.993 | −0.018 * (0.010) | 0.982 |
| Median Household Income ($1000s) | | | 0.018 *** (0.004) | 1.018 | 0.010 ** (0.004) | 1.010 |
| Vintage Housing Stock Count | | | | | 0.001 *** (0.0002) | 1.001 |
| D.C. Ward Fixed Effects | Yes | | Yes | | Yes | |
| Observations | 173 | | 173 | | 173 | |
| Pseudo $R^2$ | 0.284 | | 0.330 | | 0.397 | |

* $p < 0.1$; ** $p < 0.05$; *** $p < 0.01$.

According to both sets of models, race proves statistically significant when it is the lone independent variable (models 7.1 and 8.1). Its significance disappears when other variables are added in models 7.2, 7.3, and 8.2. It remains significant in the preferred model: NBR model 8.3. Income is statistically significant when paired with race (models 7.2 and 8.2) and when the housing vintage variable is added to create the full models (7.3 and 8.3). The housing vintage variable demonstrates strong statistical significance in both full models.

Effect size is interpreted differently in the two types of models. Consider Table 7 first because interpreting OLS regression coefficients may be more familiar to some readers than incidence rate ratios. According to the full model in Table 7 (model 7.3), holding other variables constant, a $1000 difference in median household income between two census tracts predicts 0.134 more full replacements in the wealthier tract. Thus, a $10,000

difference in median household income between two census tracts predicts 1.34 more full replacements in the wealthier tract. Comparing a census tract with the lowest median household income in our dataset ($14,572) to a census tract with the median level in the dataset ($75,703), we would expect to see an additional 8.17 full replacements in the wealthier tract. Using descriptive statistics for the dataset (see Table 3) as benchmarks for understanding the scale of these differences, we see that adding 8.17 full replacements amounts to a 51 percent increase over the mean number of fulls per census tract (15.94) or a 91 percent increase over the median number of fulls per census tract (9).

In model 7.3, the OLS regression coefficient for the control variable indicates that a census tract with one more pre-1950 housing unit than another census tract would be expected to have 0.013 more full replacements than the comparison tract, which translates to 0.08 percent of the mean census tract's full count. A census tract with 100 more pre-1950 housing units would likely have 1.3 more fulls (or 8 percent more than the mean census tract). The race variable in model 7.3 is not statistically significant and therefore cannot be used for explanation or prediction.

Negative binomial modelers (Table 8) typically convert regression coefficients into incident rate ratios (IRR), which explain the effect of each independent variable on the dependent variable in terms of impact on event probabilities or rates. Specifically, for the mean census tract, IRR measures the effect of a change in the independent variable on the ratio of the number of outcome events to the total number of times the event could have occurred, otherwise known as the exposure risk, over a specified time period. For this study, the outcome events are full LSL replacements in Washington, D.C.; the exposure risk is measured by stock of pre-1950 housing as a proxy for premises with LSLs; and the time period is 2009–2018. IRR values can be read as expected changes in event rates and are never negative. Values above 1 point to a positive relationship between the independent and dependent variables. Values below 1 indicate a negative relationship. A value of exactly 1 represents the null hypothesis.

In Table 8's full model, the coefficient on the race variable is statistically significant at the 10 percent level. The IRR indicates that having a 1 percent larger share of Black householders in a census tract predicts fewer full replacements by a factor of 0.982 (or 0.018 percent), holding other variables constant. Compare, for example, a census tract with a very low level of Black householders (1.4 percent is the lowest level in our dataset) with a census tract containing the median level of Black householders (48.3 percent). According to Model 8.3's parameters, that difference of 46.9 percentage points in racial composition would translate to a difference in full replacements between the two tracts of 0.84 percent.

The IRR for income in Model 8.3 predicts that a $1000 increase in median household income will increase that census tract's full replacement count by a factor of 1.010 or 1.0 percent. According to this parameter estimate, the difference in median household income between the lowest census tract's level ($14,572) and the median census tract's level ($75,703) predicts on average 62 percent more full replacements in the richer census tract. According to the descriptive data in Table 3, that translates to 9.9 more fulls in a census tract with the mean level of fulls for this dataset (15.94), or 5.6 more fulls in a census tract with the median level of fulls (9). The OLS estimate of 8.17 falls in the middle of the range of the NBR estimates (5.6–9.9) for the number of additional full LSLRs expected in the median-income census tract compared to the poorest census tract.

According to the IRR score for the housing vintage variable, a census tract with 1 more pre-1950 housing unit than another census tract would likely have 1.001 times more full replacements, or 0.1 percent more. This estimate is a bit higher but comes close to the OLS estimate of 0.08 percent discussed above.

### 3.4. A Note on Race and Income

Although race and income tend to be negatively correlated with each other, both variables are included in these models because they represent separate and mutually reinforcing social forces of privilege and disadvantage with potential to influence take-up

of full LSLR. Sociologists have argued that analyses of social equity need to move "beyond viewing race and economic resources as competing, rival explanations of life chances" toward an "emerging view that racial stratification causes economic inequalities earlier in and at various stages of life, and these economic inequalities then contribute to and exacerbate racial inequalities later in life" [61] (p. 599).

Table 9 reports variable inflation factors (VIF) for the independent variables in this study. Although none exceeds the threshold of 10 for determining multicollinearity, the VIF scores for percentage of Black householders are consistently higher than the others. This is not surprising given the complex social dynamics of income and race in the U.S.

**Table 9.** Variable inflation factors (VIF).

| Independent Variable | OLS Full Model 7.3 | NBR Full Model 8.3 |
|---|---|---|
| Percent Black Householders | 8.21 | 7.80 |
| Median Household Income (1000s) | 3.60 | 3.43 |
| Pre-1950 Housing Count | 1.60 | 1.57 |

Two additional model specifications were developed to further explore how Black race might mediate the role of median household income in explaining take-up of full LSLR. First, a simple multiplicative interaction term—percent Black householders x median household income in each census tract—was added to the models. Neither the interaction term nor the separate variables for race and income were statistically significant in these models. Vintage housing stock was the only statistically significant variable in these models.

Second, two-stage models were run using both OLS and NBR in an effort to isolate the effect of income apart from race. In both cases, the coefficients on income were statistically significant at the 95 percent confidence level. In the two-stage OLS model, the income coefficient was smaller than its counterpart in model 7.3. Results for the two-stage NBR model were essentially identical to the results in model 8.3, which provides additional support for the appropriateness of NBR in this study.

In light of the results from the interaction term and two-stage models and the long history of complicated interactions between race and income in the U.S., Tables 7 and 8 demand extra care in interpretation. On their face, Models 7.2, 7.3, and 8.2 suggest that income is the more powerful of the two demographic variables used in this study at the census-tract level. If true, the problem of unequal participation in full LSLR by lower-income residents may be no better or worse for households in neighborhoods with larger percentages of Black households than for others. In contrast, Model 8.3, based on the preferred method of NBR, finds both race and income to be significant explanators.

Regardless of which model comes closest to reality, it should be remembered that Black households are more likely than White households to experience lower incomes. In 2019 White residents of Washington, D.C. had a poverty rate of 5.1 percent compared to 23.4 percent for Black residents and 8.9 percent for residents of Hispanic origin, according to the American Community Survey. Thus, the equity concerns raised by this study's findings on behalf of lower-income populations are likely to have a disproportionate impact on Black households apart from whether the regression models do or do not show a statistically significant impact of race on rates of full replacement independent of income.

In addition, given the disproportionately high baseline levels of lead exposure risk experienced by Black populations in the U.S. [32] and the higher blood lead levels of Black children compared to other racial/ethnic groups [34], there are reasons to worry that risks of harm to low-income Black households from lead in drinking water may exceed risks to other low-income households. Potential for cumulative harm should be kept in mind.

As already noted, regression results at census-tract level also should be understood in the context of other area-level geographies that have shaped, and been shaped by, social forces closely entwined with race and income. In addition to the larger-area results shown in Table 6, Appendix A, and Figures 1 and 2, adding ward dummies to conventional regression

models boosts explanatory power (as measured by adjusted and pseudo R-squared values). Ward location appears to be capturing spatially distributed phenomena in the District of Columbia that are less visible at the census-tract level due to disaggregation. The political, environmental, and public health legacies of Black–White residential segregation may be one such phenomenon.

### 3.5. Geographically Weighted Regression Results

GWR models aim to address concerns about spatial autocorrelation, such as those raised by the Moran's I values we reported in Section 2.7, and thereby produce less biased regression coefficients. Tables 10 and 11 below report the average estimates from the GWR models. In each case, the GWR models offer somewhat reduced goodness of fit, as measured by adjusted and pseudo R-sqared, compared with the conventional models reported in Tables 7 and 8. This surprised the research team and suggests that the smooth distance-decay function that underlies GWR might not be fully capable of capturing the spatial realities of scio-economic and environmental justice in Washington, D.C.

**Table 10.** Association of full LSLR with race and income characteristics at census-tract level: geographically weighted OLS model results with robust standard errors (in parentheses).

| Independent Variable | Model 10 Coefficient |
|---|---|
| Percent Black Householders | 0.124 (0.078) |
| Median Household Income ($1000s) | 0.191 *** (0.051) |
| Vintage Housing Stock Count | 0.011 *** (0.003) |
| D.C. Ward Fixed Effects | Yes |
| Observations | 173 |
| Adjusted $R^2$ | 0.233 |

*** $p < 0.01$.

**Table 11.** Association of full LSLR with race and income characteristics at census-tract level: geographically weighted negative binomial regression (NBR) model results with robust standard errors (in parentheses) and incident rate ratios (IRR).

| Independent Variable | Model 11 | |
|---|---|---|
| | Coefficient | IRR |
| Percent Black Householders | 0.002 (0.005) | 1.002 |
| Median Household Income ($1000s) | 0.012 *** (0.003) | 1.012 |
| Vintage Housing Stock Count | 0.001 *** (0.0002) | 1.001 |
| D.C. Ward fixed effects | Yes | |
| Observations | 173 | |
| Pseudo $R^2$ | 0.040 | |

*** $p < 0.01$.

Beyond goodness of fit and consistent with the conventional models, median household income and housing vintage continue to be statistically significant in both GWR models. Like the conventional OLS full model (Model 7.3), the race variable is not statistically significant in GWR OLS Model 10. Unlike the conventional NBR model (Model 8.3),

where race is statistically significant (albeit weakly: at the 10 percent level), race is not statistically significant in the GWR version of NBR (Model 11).

The GWR routine allows for two hypotheses to be tested, based on using a Monte Carlo simulation of 1000 observations where the spatial points are randomly distributed amongst the data. The first test asks whether GWR describes the data better than a global regression model. This utilizes the bandwidth estimated by the routine. In all our models, we find that indeed GWR does describe the data better by this measure (but note some loss in goodness of fit, as noted above). The second test asks whether the set of parameter estimates shows substantial spatial variation. This is conducted by comparing the standard deviation of the observed parameter estimates ($S_i$) with those from the Monte Carlo simulation.

Results of the significance tests for non-stationarity are reported in Tables 12 and 13. In both tests—OLS and NBR—race is highly statistically significant. Median household income, on the other hand, achieves statistical significance only in the NBR version of GWR, and likewise for the vintage housing exposure variable. These results indicate that the role of the independent variables in explaining full LSLR probably varies from neighborhood to neighborhood. This appears to be especially likely for the race variable given its strong significance in both models.

**Table 12.** Tests of non-stationarity: GWR models using OLS.

| Dependent Variable: Count of Full LSLRs (All Types) | | |
|---|---|---|
| **Independent Variable** | **$S_i$** | ***p*-Value** |
| Constant | 56.164 | 0.509 |
| Percent Black Householders | 2.528 | 0.000 *** |
| Median Household Income ($1000s) | 0.341 | 0.888 |
| Vintage Housing Stock Count | 0.017 | 0.897 |

*** $p < 0.01$.

**Table 13.** Tests of non-stationarity: GWR models using NBR.

| Dependent Variable: Count of Full LSLRs (All Types) | | |
|---|---|---|
| **Independent Variable** | **$S_i$** | ***p*-Value** |
| Constant | 0.820 | 0.015 ** |
| Percent Black Householders | 0.015 | 0.000 *** |
| Median Household Income ($1000s) | 0.005 | 0.014 ** |
| Vintage Housing Stock Count | 0.000 | 0.055 * |

* $p < 0.1$; ** $p < 0.05$; *** $p < 0.01$.

Appendix D displays chloropleth maps based on the preferred GWR-NBR model results presented in Table 13.

## 4. Discussion

The Lead and Copper Rule (LCR)—the federal framework for regulating lead in drinking water—was promulgated in 1991 [62], one year after EPA Administrator William K. Reilly established the first-ever U.S. federal task force on socio-environmental disparities. Formation of that task force and release of its report in 1992 marked important achievements for the rapidly developing environmental justice movement and helped educate the nation about the disproportionately higher burden of environmental risk borne by low-income and minority Americans, including risk of lead exposure [63]. In 1994, President Clinton signed Executive Order 12898, which directs federal agencies, including the EPA, to implement environmental justice within their policies and programs.

Thirty years later, the revised LCR published in the Federal Register 15 January 2021 asserts that it "meets the intent" of the EJ executive order (through corrosion control

treatment) while also acknowledging a fundamental equity concern: "LSLR may not be affordable for low-income households" (86 Federal Register at 4276). As more and more states and localities embrace full LSLR as a preferred strategy for reducing the risk of lead in drinking water [17,40], the latter statement assumes greater importance in the policy conversation.

The models reported in this article provide the first available test of hypothesized affordability and equity problems in LSLR using data on actual replacements. Under cost-sharing arrangements in place during the study period, the probability of experiencing full replacement of one's LSL was significantly greater in higher-income neighborhoods of Washington, D.C. when measured at both the census tract and ward level using OLS, NBR, and GWR methods. Ward-level analysis suggests that neighborhoods with relatively smaller Black populations also benefited disproportionately from full LSLR. GWR tests for non-stationarity indicate that the relationship between race and full LSLR operates differently in different parts of the city and may interact with income differently from neighborhood to neighborhood.

The inequities identified in this study raise serious questions about which population groups have been waiting longest to receive a proper environmental remedy for the problem of lead risk in drinking water, both in Washington, D.C. and potentially nationwide. They also raise questions about how policy—specifically the national Lead and Copper Rule (LCR)—may have facilitated inequity. Although this study did not include empirical policy research, several features of the revised LCR merit discussion because of their potential to exacerbate the inequities we have identified in Washington, D.C. pre-2019.

The first is customer cost-sharing. While some of the 11,000 utilities with LSLs across the U.S. may choose to pay some or all the cost of full LSLR during planned infrastructure projects, others will not. Therefore, under the revised rule, many utilities could reasonably be expected to adopt arrangements similar to DC Water's capital improvement and customer-initiated LSLR programs. While such changes would have the positive effect of increasing total number of full LSLRs and reducing partials overall (by making full replacement easier for those willing and able to pay), our study provides evidence that wealthier customers will be more likely to participate, leaving low-income households (which are disproportionately Black) with increased risk of harm from drinking water. If so, then the unintended consequences of the policy changes could make health equity and environmental justice disparities worse, not better, than the current version of the LCR.

A second feature involves whether partial LSL replacements are allowed and if they are conducted automatically when infrastructure work is being undertaken and customers do not respond to information about full replacement. In the absence of means-tested financial assistance during the study period, Washington, D.C.'s cost-sharing-based program led to more full replacements for residents of more advantaged areas and more partial replacements for residents in more disadvantaged areas. The partial LSLRs exposed over-burdened households to additional risk based on passive rather than active consent because households that did not respond to notices provided by the utility received partial replacements by default. Such patterns translate into additional months and years of lead exposure risk for residents in disadvantaged parts of the city. Similar patterns may have occurred in cities with similar approaches to partial LSLR and customer cost-sharing: further research is needed to test that hypothesis.

The findings reported here should be considered exploratory for several reasons. First, lack of access to pre-2009 data constrained the study team's ability to fully model the distribution of full versus partial replacements across the District of Columbia. DC Water replaced several thousand LSLs prior to the study period, but those premises could not be included in the current study because the quality of the data is poor, according to utility officials. Nor could those replacements be subtracted from the counts of pre-1950 housing to ensure a more accurate control/exposure variable in the regression models because we do not know where they occurred.

Second, like many cities, Washington, D.C. has many premises with unknown LSL materials—roughly 30,000 unknowns according to DC Water. If the status of those LSLs was known, the resulting data could have been geocoded and substituted for the housing vintage data to provide a far more precise exposure variable in the regression models. In addition, with reliable data on the location of all LSLs in Washington, D.C., the models could have been run with an alternative version of the rate-type dependent variable: the ratio of full replacements to all remaining LSLs.

Third, the models do not include variables designed to capture reasons (other than income) why property owners do or do not choose to take up invitations to invest in full LSLR due to lack of empirical research on those reasons. The study team hopes researchers will expand on and refine the basic model presented here in future studies. For example, variation in the type of communications received from the utility might influence LSLR take-up, but we did not have data on that factor. Finally, as noted in Section 2.1 above, this study ideally would have utilized demographic information at the household/premise level, but this was not available.

The Washington, D.C. case study lays the foundation for future research. In addition to applying the research design described here to other cities as single case studies, we hope future studies will compare differences in full versus partial LSLR patterns between water systems operating under different policies and financing arrangements. As Washington's relatively new free and reduced-price LSLR program reaches maturity, a before-and-after study design should be applied to compare the LSLR patterns from 2009 to 2018 (reported here) with patterns being generated under the 2019 reforms.

## 5. Conclusions

This article makes two main contributions to the social science literature on lead in drinking water and related public policies. First, the article presents a versatile new indicator of environmental-remediation justice with potential utility in future analyses of LSLR programs in multiple cities. The indicator may take the form of a count variable or a ratio/rate variable, as demonstrated in our study. Either way, the core concept is the relationship between full and partial lead service line replacements and who has access to full LSLR, the superior form of remediation. The new indicator is especially relevant for jurisdictions where the lead/non-lead status of large numbers of service lines is unknown. In those contexts, researchers with access to reliable data about full versus partial replacements can use the model developed here to investigate questions of social equity within the LSLR data without having to infer underlying numbers of LSLs. This article reports results from the first such study to examine social equity dimensions of full versus partial LSLR using the new indicator.

Second, the patterns of inequity suggested by our findings have policy and program relevance far beyond one city's boundaries. Current proposed revisions to the national Lead and Copper Rule neither ban partial LSLR nor facilitate funding for full replacement. In the absence of further revisions to the LCR, key features of DC Water's LSLR programs during the study period are likely to persist in many other water systems. Of specific concern from a social-equity perspective are customer cost-sharing requirements for all full replacements and program designs that authorize partial replacement as a utility's default action when customers do not respond to opportunities for full replacement. Further research is needed to test the robustness of the Washington, D.C. findings in other jurisdictions and under other program settings.

Further research is also needed before specific, evidence-based policy remedies can be recommended to address LSLR inequities. In the meantime, the Washington, D.C. case study points to areas for potential improvement. Washington, D.C.'s free-and-reduced-cost LSLR policy is designed to directly address the issues of environmental injustice and health inequity documented in this article. As of October 2019, the city began fully covering the cost of every full LSLR associated with new infrastructure work for all customers. The policy also offers generous subsidies for full LSLR at premises where partial replacements were

previously conducted (covering 80–100 percent of the cost for families with incomes below the area median income). The policy includes additional outreach and public education initiatives with the goal of increasing demand for full replacements to correct past partials. Other cities are designing and implementing additional innovations to address economic barriers to full LSLR [40].

Funds on a scale far beyond existing grant and loan programs are needed to extend these types of remedies to all U.S. towns and cities with LSLs in need of replacement. Several infrastructure bills introduced in the U.S. Congress in 2021 contain funding that would significantly accelerate the national pace of LSLR and thereby reduce the risks for those who must wait the longest for full replacement. The amount of funding approved for LSLR by the U.S. Congress in November 2021 is roughly one-third of the original proposals, which were based on estimates of total funding needed.

In March 2021, the EPA extended the effective date of the revised LCR to allow additional public input, "particularly from individuals and communities that are most at risk of exposure to lead in drinking water." [64]. On 16 December 2021, EPA allowed the revised rule to go into effect, but committed to starting the process to propose a new rule that will incorporate principles of 100 percent lead service line replacement and improving public health protection for those who cannot afford to replace the customer-owned portion of the line [30]. Among the latter, EPA identifies the need to prioritize "historically underserved communities" and those "disproportionately impacted by lead in drinking water" [65] (p. 2). This article offers federal policy makers important clues about the potential unintended consequences of current policy for these priority populations.

**Author Contributions:** Conceptualization, K.J.B., L.M. and T.N.; methodology, K.J.B., M.M., M.J.A., L.M. and T.N.; software, M.J.A. and R.H.; validation, M.M., M.J.A. and R.H.; formal analysis, M.M., M.J.A. and R.H.; investigation, K.J.B., M.M., M.J.A. and R.H.; resources, L.M. and T.N.; data curation, K.J.B., M.J.A. and R.H.; writing—original draft preparation, K.J.B. and M.J.A.; writing—review and editing, all authors; visualization, M.J.A., R.H., L.M. and T.N.; supervision, K.J.B.; project administration, K.J.B. and T.N.; funding acquisition, T.N. All authors have read and agreed to the published version of the manuscript.

**Funding:** The authors gratefully acknowledge funding from the Robert Wood Johnson Foundation Grant I.D. No. 76138 via the Environmental Defense Fund.

**Institutional Review Board Statement:** The study was approved by the Institutional Review Board of American University in Washington, D.C. (Protocol #: IRB-2019-258) on 15 February 2019.

**Informed Consent Statement:** Not applicable.

**Data Availability Statement:** Administrative data used in this study was made available under a memorandum of agreement with DC Water that does not allow public sharing.

**Acknowledgments:** Authors wish to thank DC Water for providing access to LSLR data and for answering our many questions. The Center for Environmental Policy (CEP) at American University provided valuable support throughout the project: special thanks go to CEP Director Dan Fiorino and John Reeder. Earlier versions of this research benefitted from research assistance by Carly Weted and Theo Affonso Laguna. Three anonymous reviewers and the editors of the special issue offered many helpful comments that greatly improved the article. We are especially grateful for the editors' suggestion to apply geographically weighted regression.

**Conflicts of Interest:** The authors declare no conflict of interest. American University and the Environmental Defense Fund are customers of DC Water. The Environmental Defense Fund (EDF) provided funding to American University to support the work: two of its employees are coauthors. EDF's funder for the work had no role in the design of the study; the collection, analyses, or interpretation of data; the writing of the manuscript; or the decision to publish the results.

## Appendix A. Bivariate Correlations at Ward Level

**Table A1.** Rates of full LSLR as a percentage of all LSLRs correlated with demographic characteristics at ward level, by program, Washington, DC, 2009–2018.

| | Bivariate Correlation with Percent Full | | |
| | | Utility-Initiated | |
| Demographic Characteristic | All LSLR Customer-Initiated + Utility-Initiated | Capital Improvement Projects | Emergency Repairs |
|---|---|---|---|
| Percent African American/Black Householders | −0.702 * | −0.781 ** | 0.345 |
| Median Household Income | 0.810 ** | 0.820 ** | −0.539 |
| Social Deprivation Index (SDI) † | −0.838 *** | −0.921 *** | 0.596 |

* Statistically significant at 10 percent level: critical r = |0.621|; ** Statistically significant at 5 percent level: critical r = |0.707|; *** Statistically significant at 1 percent level: critical r = |0.834| † SDI is a rank-order index based on a cluster of 7 social background variables often associated with poor health and education outcomes [66]. The components of SDI are percentages of area residents with the following characteristics: income below poverty, single-parent household, living in rental housing, living in overcrowded housing, education less than 12 years, lacking a car, and experiencing non-employment. Each census tract's score represents its centile ranking (scale of 0–100). Higher index numbers indicate more deprivation. SDI is expected to be correlated negatively with full replacements because it captures multiple sources of social disadvantage.

## Appendix B. Alternative OLS Model

Log transformation offers another approach to handling skewed variables. In addition, running the dependent and independent variables as logs provides direct estimates of the elasticities of the independent variables [67]. In the case of full LSLR, we are especially interested in estimating the neighborhood equivalent of income elasticity of demand for full LSLR. The table below reports OLS results for log-transformed versions of all variables. These models, and those in Appendix C, use the Black population count in each census tract rather than percentage Black householders because of the challenges of interpreting a percent variable that has been logged.

**Table A2.** Total full LSLRs modeled as a function of median household income, Black population, and vintage housing stock, with all variables logged, by census tract, Washington, DC, 2009–2018, using OLS (robust standard errors in parentheses).

| Dependent Variable: Logged Count of Full LSLRs, All Types | | | |
|---|---|---|---|
| **Independent Variable** | **Model B1.1** | **Model B1.2** | **Model B1.3** |
| Log Percent Black Householders | −0.268 ** (0.134) | −0.081 (0.147) | −0.048 (0.141) |
| Log Median Household Income ($1000s) | | 0.884 *** (0.322) | 0.712 ** (0.312) |
| Log Vintage Housing Stock Count | | | 0.667 *** (0.194) |
| D.C. Ward fixed effects | Yes | Yes | Yes |
| Observations | 129 | 129 | 129 |
| Adjusted $R^2$ | 0.352 | 0.385 | 0.436 |

** $p < 0.05$; *** $p < 0.01$.

The coefficients on income in the preferred models (with fixed effects) suggest that full replacement of a lead service line functions as what economists call a normal good at the neighborhood level, meaning demand for full LSLR has an income elasticity between 0 and 1 [68]. The coefficients in those models show percentage increases (decreases) in de-

mand for full LSLR to be positive and less than or equal to percentage increases (decreases) in income.

**Appendix C. Program Effects**

We also ran the log-transformed models separately by program to compare income elasticities. The table below reports OLS results for all full customer-initiated (CI) LSLRs by census tract with all variables logged.

**Table A3.** Customer-initiated (CI) full LSLRs modeled as a function of median household income, Black population, and vintage housing stock, with all variables logged, by census tract, Washington, DC, 2009–2018.

| Independent Variable | Dependent Variable: Logged Count of Full LSLRs, Customer-Initiated Only | | |
|---|---|---|---|
| | Model C1.1 | Model C1.2 | Model C1.3 |
| Log Percent Black Householders | −0.093 (0.130) | 0.130 (0.141) | 0.143 (0.135) |
| Log Median Household Income ($1000s) | | 1.033 *** (0.309) | 0.863 *** (0.300) |
| Log Vintage Housing Stock Count | | | 0.623 *** (0.189) |
| D.C. Ward fixed effects | Yes | Yes | Yes |
| Observations | 117 | 117 | 117 |
| Adjusted $R^2$ | 0.348 | 0.404 | 0.454 |

*** $p < 0.01$.

**Table A4.** Utility-initiated (UI) full LSLRs modeled as a function of median household income, Black population, and vintage housing stock, with all variables logged, by census tract, Washington, D.C., 2009–2018, using OLS (robust standard errors in parentheses).

| Independent Variable | Dependent Variable: Logged Count of Full LSLRs, Utility-Initiated Only (Includes Capital Improvement-Related and Emergency-Related UI) | | |
|---|---|---|---|
| | Model C2.1 | Model C2.2 | Model C2.3 |
| Log Percent Black Householders | −0.260 (0.186) | −0.110 (0.192) | −0.071 (0.190) |
| Log Median Household Income ($1000s) | | 0.758 * (0.390) | 0.616 (0.392) |
| Log Vintage Housing Stock Count | | | 0.518 * (0.267) |
| D.C. Ward fixed effects | Yes | Yes | Yes |
| Observations | 96 | 96 | 96 |
| Adjusted $R^2$ | 0.108 | 0.131 | 0.150 |

* $p < 0.1$

More severe inequities of outcome might be expected in the distribution of CI replacements (all of which are full) compared with utility-initiated LSLRs for two reasons. First, as noted previously, CIs are 28 percent more expensive on average than UI fulls. That steeper price is likely to discourage lower-income property owners more than their higher-income counterparts. In addition, CI replacements require more pro-active effort on the homeowner's part than UIs. Thus, residents from more privileged social groups may feel more confident reaching out to DC Water and initiating the work. Full replacements were expected to occur more often in higher-income, more privileged neighborhoods under both the UI and CI programs, but moreso in the latter for those reasons.

While race is not statistically significant in any of the logged models in either table, the log of median household income is significant in models C1.2, C1.3, and C2.2, and it comes close in model C2.3 ($p = 0.119$). These results provide some support for the program-effect hypothesis while also illustrating weakness in the Table A4 models for predicting full utility-initiated LSLR.

### Appendix D. Chloropleth Maps

GWR tests of non-stationarity suggest that race, income, and take-up of full LSLR opportunities may interact in different ways in different parts of Washington, D.C., as reported in Tables 12 and 13 above. Figures A1 and A2 below allow further exploration of these differences based on the results of the preferred model specification using NBR.

According to Figure A1, the race variable (percentage of Black householders) behaves mostly as expected in parts of the city with the largest percentages of Black households: namely, Wards 5, 7, and 8. GWR models centered in the census tracts comprising those wards (plus part of Ward 6) generate mostly negative coefficients for the race variable and those negative coefficients get larger the farther east one goes toward the city's more concentrated Black neighborhoods. Moving west through mixed neighborhoods and toward the whiter areas of the city, coefficients on the race variable are closer to zero, suggesting that race has less explanatory power in those areas. Some of the coefficients in the farthest west neighborhoods have positive signs and larger effects sizes—an unexpected result that deserve attention in further research.

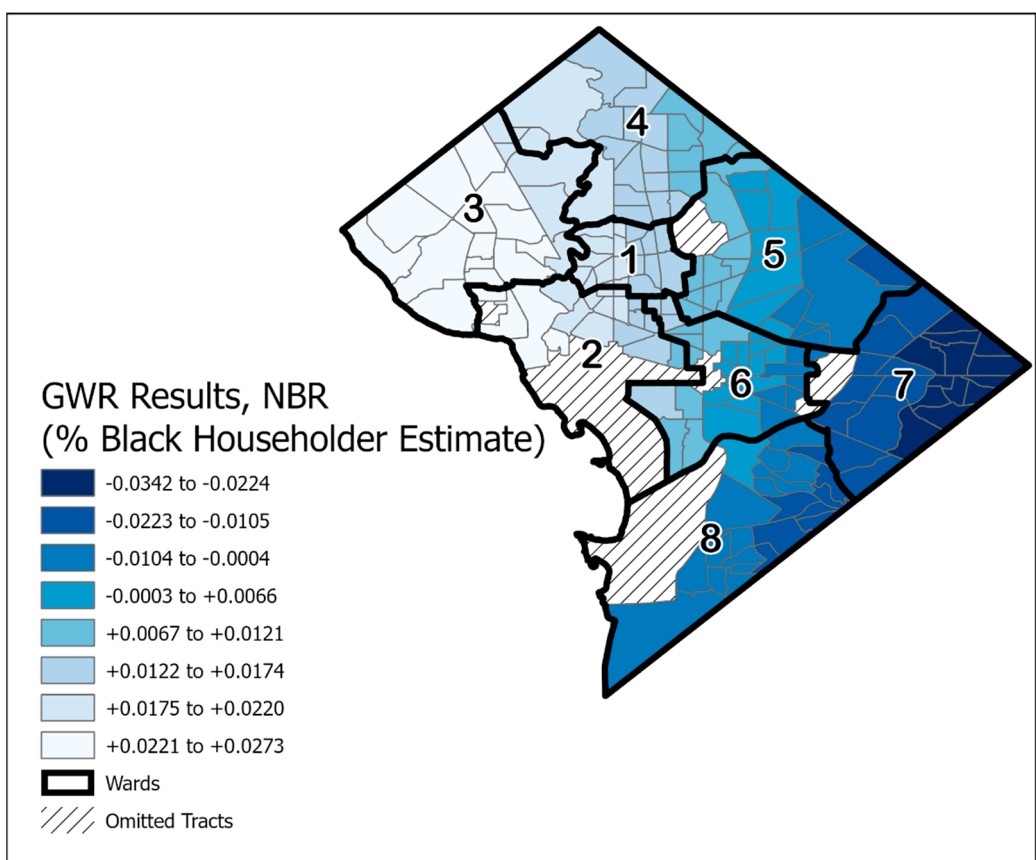

**Figure A1.** Distribution of race coefficient estimates for GWR models by census tract using NBR specification.

According to Figure A2, the income variable (median household income) behaves mostly as expected in parts of the city with the largest incomes: namely, Wards 1, 2, and 3, and parts of 4 and 6. GWR models centered in the census tracts comprising those wards generate positive coefficients for the income variable and those positive coefficients get

larger the closer one moves toward the heart of downtown D.C., roughly opposite Wards 7 and 8 with their dramatically lower median incomes. In the far eastern census tracts, the signs on the income coefficients are very close to zero (and some are actually negative), perhaps because residents of those neighborhoods have more uniformly low incomes, which makes differences in income a less useful predictor of full LSLR.

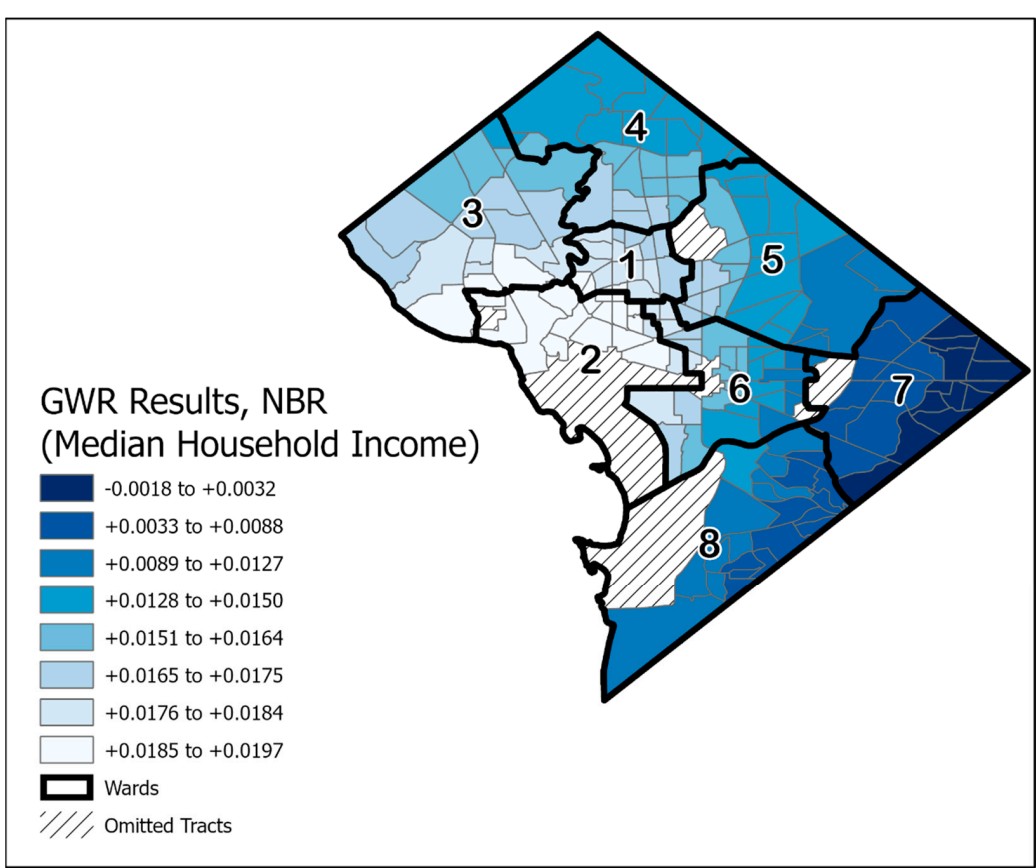

**Figure A2.** Distribution of income coefficient estimates for GWR models by census tract using NBR specification.

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
