# Peer review of "Full Lead Service Line Replacement: A Case Study of Equity in Environmental Remediation"

_sustainability, doi:10.3390/su14010352_

Round 1

Reviewer 1 Report

Introduction:

Currently the paper is too weak in terms of framing the paper with respects to other research, with respects to the method of data collection and with respects to the significance for readers. I see nothing on the health effects of lead exposure which is the primary reason for removing LSLs.

The article needs to  mention and cite some of the major systems it affects like-

1)      cardiovascular system

Lanphear, Bruce P., Stephen Rauch, Peggy Auinger, Ryan W. Allen, and Richard W. Hornung. "Low-level lead exposure and mortality in US adults: a population-based cohort study." The Lancet Public Health 3, no. 4 (2018): e177-e184.

Obeng-Gyasi, E., Ferguson, A.C., Stamatakis, K.A. and Province, M.A., 2021. Combined Effect of Lead Exposure and Allostatic Load on Cardiovascular Disease Mortality—A Preliminary Study. International journal of environmental research and public health18(13), p.6879.2)      

2) Renal system

 Harari, Florencia, Gerd Sallsten, Anders Christensson, Marinka Petkovic, Bo Hedblad, Niklas Forsgard, Olle Melander et al. "Blood Lead Levels and Decreased Kidney Function in a Population-Based Cohort." American Journal of Kidney Diseases (2018).

 Lin, Ja-Liang, Dan-Tzu Lin-Tan, Kuang-Hung Hsu, and Chun-Chen Yu. "Environmental lead exposure and progression of chronic renal diseases in patients without diabetes." New England Journal of Medicine 348, no. 4 (2003): 277-286.

3)      Hepatic system

Can, S., C. BaÄŸci, M. Ozaslan, A. I. Bozkurt, B. Cengiz, E. A. Cakmak, R. KocabaÅŸ, E. KaradaÄŸ, and M. TarakçioÄŸlu. "Occupational lead exposure effect on liver functions and biochemical parameters." Acta Physiologica Hungarica 95, no. 4 (2008): 395-403.

 etc 

 Methods:

  • A full and transparent justification of sample sizes are needed for all analysis conducted.
  • Did you consider the effects of clustering at all levels?
  • Was a methodologist consulted prior to this study?

Results:

Put error bars on all bar charts

Why are no covariates adjusted for in any of the analysis?

Discussion:

The discussion does not give enough context to the results from other works and from the findings. More is needed.

Conclusion:

There needs to be more consistency between the conclusions and the discussion

Reviewer 2 Report

This manuscript presents a detailed analysis of the influence of selected socioeconomic factors for predicting if lead service lines would be replaced fully (from main to customer) or partially (only the public property section). The study method is well suited to the problem and was executed appropriately. The outcomes of the study will serve as an example of how to approach understanding other social equity issues in drinking water and public health. 

Comments for consideration

Introduction

  • for readers which may not be familiar with the terminology an explanation of what a lead service line is required
  • The conclusions focus a lot on the differences found in the Black community which is explained in section 3.4. It would be beneficial to provide a clearer explanation of this in the introduction and supported in the literature review. Especially how the black community compare with other minorities associated with lower socioeconomic outcomes. 
  • Line 104: given the importance of equity issues some detail should be provided by other approaches and how further research is required to overcome current limitations
  • Lines 111 -115 - this is an example of some of the long sentences, consider shortening long sentences or breaking into multiple sentences. 

Materials and methods

  • following on from the comment on race in the introduction, a clearer explanation for selecting income and race (in particularly selecting black over all racial groups) as predictors is required

Results

  • Remove the template wording at lines351-353

Discussion and Conclusions

- The wording in the conclusion is largely repitious of the discussion content. The authors should consider implications for public health policy, health regulators etc in the conclusions 

Reviewer 3 Report

This paper seeks to explore the effect of neighborhood-level, socioeconomic characteristics on the rate of full service line replacements in Washington D.C. from 2009 to 2018 to understand if lead service line mitigation efforts are equitable. This is a well-written paper that provides important policy insights for the D.C. water utility and others addressing lead service lines around the country. I recommend this paper be accepted with minor revisions. My primary concern with the paper is related to the authors' treatment of the race variable in the models (i.e., percent Black householders). My recommendation to the authors would be to consider testing interaction terms between race and income in their existing models and/or test a two-stage regression model to more accurately evaluate the role of race and how it relates to income and, consequently, LSLs. More detailed specific comments are included in the attached document. Thank you for the opportunity to review this important work. 

Round 2

Reviewer 1 Report

The manuscript is much improved and ready for publication.

Author Response

Thank you very much for this second positive review. We appreciate your attention to our manuscript.

Regards,

The Authors